# The Transcribed-Ultra Conserved Regions: Novel Non-Coding RNA Players in Neuroblastoma Progression

**DOI:** 10.3390/ncrna5020039

**Published:** 2019-06-04

**Authors:** Nithya Mudgapalli, Brianna P. Shaw, Srinivas Chava, Kishore B. Challagundla

**Affiliations:** 1UNMC Summer Undergraduate Research Program, Omaha, NE 68198, USA; nmudgapalli@gmail.com (N.M.); brianna.shaw@unmc.edu (B.P.S.); 2Department of Biochemistry and Molecular Biology & The Fred and Pamela Buffett Cancer Center, University of Nebraska Medical Center, Omaha, NE 68198, USA; srinivas.chava@unmc.edu

**Keywords:** transcribed-ultra conserved regions, microRNAs, MYCN, exosomes, metastasis, biomarkers, oncogenes, therapy resistance, neuroblastoma

## Abstract

The Transcribed-Ultra Conserved Regions (T-UCRs) are a class of novel non-coding RNAs that arise from the dark matter of the genome. T-UCRs are highly conserved between mouse, rat, and human genomes, which might indicate a definitive role for these elements in health and disease. The growing body of evidence suggests that T-UCRs contribute to oncogenic pathways. Neuroblastoma is a type of childhood cancer that is challenging to treat. The role of non-coding RNAs in the pathogenesis of neuroblastoma, in particular for cancer development, progression, and therapy resistance, has been documented. Exosmic non-coding RNAs are also involved in shaping the biology of the tumor microenvironment in neuroblastoma. In recent years, the involvement of T-UCRs in a wide variety of pathways in neuroblastoma has been discovered. Here, we present an overview of the involvement of T-UCRs in various cellular pathways, such as DNA damage response, proliferation, chemotherapy response, *MYCN* (*v-myc myelocytomatosis viral related oncogene, neuroblastoma derived* (*avian*)) amplification, gene copy number, and immune response, as well as correlate it to patient survival in neuroblastoma.

## 1. Neuroblastoma

Neuroblastoma is a type of peripheral sympathetic nervous system cancer, affecting mostly infants and young children (95% of which are under the age of 5, and occurring 13% more frequently in males), which alters the growth and proliferation of neural crest cells (precursor nervous system cells) [1]. Neuroblastoma has a diverse clinical response to current treatments across the patient population and is quite rare, making research difficult. In Europe, alone, the annual incidence rate is recorded to be six cases/million [2]. Some children respond well to treatment and eventually are deemed cured of their cancer, while some children’s cancer spontaneously regresses on its own, but others develop a strong resistance to treatments and a poor prognosis remains [3,4,5,6]. Infants have the best prognosis of all age groups, with a 5-year survival rate of 91%. However, the 5-year survival rate decreases with age of onset, with a 56% survival rate for children 10–14 years of age [7,8]. Most often, neuroblastoma originates within either the adrenal glands, the paravertebral ganglia, and/or the neck as a solid tumor, and can potentially spread, although it is normally caught prior to widespread malignancy [2]. Surgery, chemotherapy, and radiotherapy are all current options for treatment depending upon the characteristics of the tumor’s presentation and behavior; however, chemotherapy is currently the main treatment option [1]. It was discovered that an amplification of the *MYCN* (*v-myc myelocytomatosis viral related oncogene, neuroblastoma derived* (*avian*)) gene in Neuroblastoma patients was correlated with an increased aggressive behavior of the tumor, leading to a poor prognosis [9,10]. Currently, research is focused on new therapies aimed at attempting to target and inhibit both the MYCN amplification process, as well as the tumorigenesis of the cancer [11].

## 2. Transcribed-Ultra Conserved Regions

Bezarano et al. were the first to discover ultra-conserved regions (UCRs), using a bioinformatics approach from the genome [11]. UCRs are 481 elements longer than the 200 base pairs that are 100% conserved, without any deletions or insertions, between the orthologues regions of human, rat, and mouse genomes. Protein-coding genes represent anywhere from 1% to 2% of the human genome; therefore, the scientific community was ignoring the rest of 98% of the genome, referred to as “junk DNA.” Recently, a group led by Axel Visel described the functional role of non-coding DNA elements in mice. Authors used genome editing technology to create knockout mice lacking individual or a group of ultra-conserved elements. Mice with deletions of ultra-conserved elements showed neurological abnormalities, including structural brain defects [12]. UCRs represent a small portion of the “junk DNA” and are likely to be involved in different biological pathways. Based on their localization, UCRs are classified into five groups: exonic, partly exonic, exon-containing, intronic, and intergenic, as shown in Figure 1 [13].

The transcripts that are transcribed from UCRs are called Transcribed-Ultra Conserved Regions (T-UCRs), which can either be “sense” (transcribed in the same orientation) or “anti-sense” (transcribed in the opposite direction). If the T-UCRs are transcribed towards the host gene, they are called sense direction, whereas they are called anti-sense direction in the opposite direction of the host gene. T-UCRs are defined as a novel class of long, non-coding RNAs. Furthermore, the presence of cancer-specific mutations in UCRs raises the question of their potential role in cancer biology [14]. In addition, UCRs are located within cancer-associated genomic regions, suggesting a role in cancer biology [15].

## 3. Regulation of Transcribed-Ultra Conserved Regions

Calin et al. were the first group to describe the differential expression of T-UCRs in human cancers, specifically in chronic lymphocytic leukemia (CLL; a cancer type of the blood and bone marrow), colorectal carcinoma (cancer of the colon or rectum), and hepatocellular carcinoma (primary malignancy of the liver) models [16]. These studies were performed in vitro, using patient samples or cell line models, as well as in silico, using various bioinformatics tools. The authors concluded that cancer cells express distinctive T-UCR and miRNA signatures when compared to their respective controls. Based on the in vitro experiments, alterations in T-UCRs and miRNAs indicated that they regulate each other, suggesting that the coding and non-coding genes cooperate to play a vital role in the biology of malignancy. Regulation of T-UCRs in various cancers has been found in four potential ways, including altered interactions with miRNAs, hypermethylation of CpG islands at the promoter region of the protein-coding gene [17], trimethylation of histone H3 (H3K4me3) located near the transcription site of the protein-coding gene, and hypoxia [13]. These data were obtained from 59 cancer cell lines and 283 primary tumors treated with or without 5-aza-2′-deoxycytidine, a demethylation agent [17].

### 3.1. Regulation of Gene Expression by T-UCRs

Recent evidence highlights the importance of T-UCRs in the regulation of gene expression through direct interaction with mRNA [18,19]. Wang et al. performed a series of experiments using colorectal cancer tissue and cell lines, exploring the possible connection between uc.338 and tissue inhibitor of metalloproteinase-1 (TIMP-1) mRNA in a model of colorectal cancer. The authors discovered that uc.338 negatively regulates TIMP-1 levels through direct interaction with the 3′UTR of TIMP-1 mRNA, suggesting that uc.338 acts as an oncogene [18] (Figure 2A). The authors used 293T and colorectal cancer cell lines (SW480 and HCT116) in the studies. This was the first study to demonstrate the function of T-UCRs in negative regulation of mRNA by direct interaction at 3′UTR [18]. A similar mechanism has been found in non-small-cell lung cancer involving the negative regulation of Heat shock protein family A member 12B (HSPA12B) mRNA by uc.454 through direct interaction at the 3′UTR of HSPA12B mRNA [19]. Moreover, uc.8 acts as a decoy molecule through the direct binding of miRNA-596, suggesting a new regulatory loop between T-UCRs and miRNAs [20]. The authors have used bladder cancer tissues and bladder cancer cell lines, and have performed experimental techniques, such as miR-binding domain accessibility, RNA binding affinity, and RNA species abundance, to find that uc.8 translocates from the nucleus to the cytoplasm in this decoy mechanism.

A similar mechanism exists in lead-induced neurotoxicity by uc.173 through degradation of pro-apoptotic miRNA-291a-3p. The authors delineated the mechanistic aspects of the regulation of miRNA-291a-3p by uc.173 by using blood samples from 200 students who were living in the area of lead-associated plants, as well as studies on N2a mouse nerve cell lines [21]. High uc.173 expressions were positively correlated with lead concentrations in the blood, thus providing a new mechanism of lead-mediated nerve damage (Figure 2B). A group led by Manel Esteller et al. discovered a novel interaction between the T-UCR named uc.283A and the stem region of the pri-miR-195 transcript through complementary base pairing, which prevents the inhibition of pri-miRNA processing by the Drosha/DGCR8 microprocessor complex [22]. A mutated version of pri-miR-195 uncouples uc.283A-mediated regulation, in both in vivo and in vitro systems. The authors revealed a novel model for the regulation of miRNA through inhibiting microprocessor-mediated recognition and cleavage of primary-miRNA [22] (Figure 2C,D).

### 3.2. Regulation of T-UCR Expression

Some of the miRNAs show significant complementarity with T-UCRs. Additionally, expression of T-UCRs was negatively correlated with the appearance of miRNAs. Experimental evidence indicates that miRNA-155 directly targets uc.160, whereas miRNA-24-1 and miRNA-29b downregulate uc.346A and uc.348 in MEG01 leukemia cells [16]. Furthermore, miRNA-153 suppresses uc.416 through direct binding in a model of gastric cancer [23]. Similarly, five miRNAs regulate the expression of nine T-UCRs in neuroblastoma [24] while, in gastric cancer, miRNA-153 acts as a tumor suppressor by directly downregulating the expression of uc.416 [23], suggesting that miRNAs may regulate the expression of T-UCRs (Figure 3A). These results were concluded from experiments involving CLL patient samples [16], gastric cancer cell lines and patient tissue samples [23], and neuroblastoma patient tumor tissue samples [24].

Hypermethylation regulates the expression of protein-coding genes and miRNAs. To test whether hypermethylation of CpG islands also regulates the expression of T- UCRs, Lujambio et al. screened the CpG island methylation status within a 2000-bp region upstream of the sense transcripts and found that three T-UCRs (uc.160, uc.283A, and uc.346) correlated with methylation status of CpG island promoter regions, suggesting a link between hypermethylation and T-UCR silencing [25,26]. Another group, led by Yasui W, also found methylation-mediated downregulation of the T-UCR, uc.158 [23]. It has also been reported that treatment with 5-AzaC, a nucleoside-based DNA methyltransferase inhibitor, restored expression of T-UCRs in HCT116 (a colon cancer cell line) and LNCaP (a prostate cancer cell line) [17,23,27], suggesting that epigenetic mechanisms regulate the expression of T-UCRs (Figure 3B).

To understand the regulation of T-UCRs transcription, Mestdagh and colleagues explored the distant distribution of the trimethylation status of lysine 4 molecule on histone H3 protein (H3K4me3), a mark for transcriptional initiation for intergenic T-UCRs, intragenic T-UCRs, and protein-coding genes, using four different neuroblastoma cell lines. This group found that intergenic and intragenic T-UCRs significantly associate with active H3K4me3, but not with protein-coding genes, which suggests that a different transcriptional regulation mechanism may exist between intergenic T-UCRs, intragenic T-UCRs, and protein-coding genes [13] (Figure 3C). Furthermore, miRNAs were also associated with the trimethylation status of H3K4me3, suggesting a common mechanism of regulation in these two non-coding RNA classes [28,29,30]. A recent study by Ferdin J. and colleagues found that hypoxia induces upregulation of several T-UCRs (uc.63, uc.73, uc.106, uc.134, and uc.475), named the ‘hypoxia-induced noncoding ultra-conserved transcripts’ (HINCUTs), partly through hypoxia-inducible factor 1-alpha (HIF1A) [31]. The authors performed a series of experiments involving cell lines from colon cancer, breast cancer, bladder cancer, and glioblastoma, with or without hypoxic conditions, thus providing the first evidence of the functional network between hypoxia and T-UCRs [31].

## 4. Transcribed-Ultra Conserved Regions in Neuroblastoma

### 4.1. T-UCRs Expression and Patient Survival in Neuroblastoma

The team involving Scaruffi, P et al. was the first to analyze the deregulation of the microRNA/T-UCR network and its correlation to survivorship in neuroblastoma. The team collected 34 tumor specimens from stage 4, high-risk neuroblastoma patients diagnosed between 1990 and 2006 at the Gaslini Children Hospital in Genoa, Italy. Patients were divided into short- and long-term survivors. Short term survivors were patients who did not survive with the disease longer than 36 months after diagnosis, while long-term survivors were patients who lived longer than 36 months after diagnosis. All patients were over the age of one-year-old. The study sought to compare the T-UCR expression levels of short- vs. long-term survivors to create a “T-UCR threshold risk-prediction model.” They analyzed the expression of all 481 UCRs, and their associated 723 microRNAs, using qRT-PCR and microarray analysis, respectively. Of the T-UCRs examined, 460 were detectable and, of these, 54 were differentially expressed between the short- and long-term survivor groups. Interestingly, the expression levels of nine T-UCRs were inversely correlated with the five microRNA signatures that have complementary binding sites, indicating a negative regulation of T-UCRs by direct interaction with microRNAs (Table 1).

The authors have concluded that complementary microRNA down-regulation may cause the up-regulation of T-UCRs within short-term survivors in neuroblastoma patients [24]. This was the first report describing the association of T-UCRs with survival in neuroblastoma patients. Nevertheless, the authors did not attempt to study the gene expression profiles and their correlation with T-UCR signatures and their outcomes in both long-term and short-term survivors of neuroblastoma patients.

### 4.2. T-UCR Expression, Genomic Locations, and Coding Genes in Neuroblastoma Tumors

For the first time, the research team of Mestdagh et al. used a functional genomic approach to designate functions to each of the T-UCRs [13]. The authors collected 49 neuroblastoma tumors from the Ghent University Hospital in Ghent (Belgium), the Medical School of Valencia in Valencia (Spain), and an additional cohort of 366 neuroblastoma tumors, as described [32]. The staging was confirmed according to the International Neuroblastoma Staging System [6]. Using a qRT-PCR-based approach, the research team analyzed the expression of all 481 T-UCRs in neuroblastoma tumors with respect to genomic location. They then correlated these findings with host and surrounding genes in neuroblastoma patients and validated the results using a cellular model system. These T-UCRs were found to be categorized as intronic (42.6%), intergenic (38.7%), exon containing (5.6%), partly exonic (5%), and exonic (4.2%). To find out whether intragenic T-UCRs independently express their host protein-coding genes, the authors also quantified T-UCR and mRNA levels (by RT– qPCR (*n* = 49) and exon array (*n* = 40), respectively) and correlated the results to their host gene expression in neuroblastoma tumors. The authors found that almost half (237) of the T-UCRs (inter and intragenic) are expressed independently of their host gene expression, while 17 T-UCRs (inter and intragenic) showed a negative correlation to one of the flanking up- or downstream-coding genes. Otherwise, none of the T- UCRs (intragenic) showed a negative correlation to their host gene.

### 4.3. T-UCR Expression and Histone Marks in Neuroblastoma

In studying the initiation and regulation of T-UCR transcription, the genomic neighborhood surrounding T-UCRs was analyzed to understand the chromatin state (specifically, the trimethylation of lysine 4 of histone H3 (H3K4me3) that indicates active transcription) in neuroblastoma patient samples and cell lines [13]. Results demonstrated that both intergenic and intragenic T-UCRs were associated with active H3K4me3 marks, but with a different distribution when compared with protein-coding genes, documenting a different transcriptional regulation between T-UCRs and protein-coding genes. These results are in agreement with other reports [30,33,34].

### 4.4. T-UCR Expression and MYCN Amplification in Neuroblastoma

The obtained T-UCR data was compared with a clinical and genetic subgroup of the patients to see if T-UCRs have any prognostic value in neuroblastoma. An upregulation of seven T-UCR signatures (four intergenic, three intronic; uc.279, uc.347, uc.350, uc.364, uc.379, uc.446, and uc.460) was found in MYCN-amplified tumors (*n* = 18) as compared to MYCN-non-amplified tumors (*n* = 31). Interestingly, none of the T-UCRs were downregulated in the MYCN-amplified tumors. Out of the seven T-UCRs, three were randomly selected (uc.279, uc.364, and uc.460) to be evaluated for expression in a neuroblastoma cohort (*n* = 366), which led to finding significant upregulation in two of the T-UCRs (uc.279, uc.460) in MYCN-amplified tumors. The seven T-UCR signatures were also validated using a SHEP-MYCN-ER cell line (4-hydroxy tamoxifen-induced MYN activation cell line) [35], which found an upregulation (more than two-fold) in three of the seven T-UCRs (uc.350, uc.379, and uc.460), suggesting that MYCN induces the expression of these T-UCRs.

### 4.5. T-UCR Expression & DNA Copy Number in Neuroblastoma

DNA copy-number affects gene expression in cancers. Chromosomal abnormalities, such as deletions of 1p, 3p, and 11q, as well as the gain of 17q, have been shown to positively influence the progression of neuroblastoma [36,37,38,39]. A team led by Mestdagh et al. tested to see if the expression of any T-UCRs correlates with DNA copy-number changes [13,39,40]. The authors identified a seven T-UCR signature that correlated with DNA copy-number (Table 2). These findings established that T-UCR deregulation associates with DNA copy number changes in neuroblastoma tumors. However, the exact manner in which T-UCR signatures influence DNA copy number remains unknown in the context of neuroblastoma.

### 4.6. T-UCR Expression and p53 Response in Neuroblastoma

In identifying putative functions of T-UCRs, the authors followed a functional genomics approach to find that T-UCRs were correlated with the function of protein-coding genes in neuroblastoma tumors [13]. A Gene Set Enrichment Analysis (GSEA) was performed to further inquire into which pathways T-UCRs are potentially involved in, which demonstrated an association with various pathways related to cancer cell proliferation, cell cycle, apoptosis, DNA repair, and differentiation [41]. Additionally, the authors also validated the involvement of T-UCRs in p53 activation using neuroblastoma cells following the inhibition of p53 with the lentiviral shRNA system and treatment with nutlin, a small molecule that activates p53 by inhibiting the interaction between MDM2 and p53 [42,43], which showed that 29 out of 40 T-UCRs are p53 responsive. Thus, these findings suggest that T-UCRs mediate the p53 responsive pathway. These results were inconsistent with other investigators, who reported that uc.73 is p53 responsive and plays a role in inducing apoptosis [16].

### 4.7. T-UCR Expression Network in Neuroblastoma

Bejerano et al. first discovered 481 T-UCRs that are conserved and located within or around the genes involved in various functional properties [11]. From these findings, a network approach was used to find pairwise correlations of 237 independently expressed T-UCRs in neuroblastoma tumors [13], of which four major clusters of T-UCRs were identified. Functions were then assigned to each cluster by GSEA. Each cluster had a function that is closely associated with common cancer relation pathogenesis: Cluster 1 consisting of nine T-UCRs was associated with DNA damage response (TP53 responsive); Cluster 2, comprised of 11 T-UCRs, was associated with cell cycle regulation and proliferation; Cluster 3, containing nine T-UCRs, was associated with p53-dependent neuronal differentiation; and Cluster 4, involving six T-UCRs, was associated with immune response and development. The correlation of these Clusters with patient survival was analyzed, which showed that Cluster 4 was significantly correlated with overall and event-free patient survival, suggesting a possible predictor of patient prognosis (Table 3). In all, these results indicated that T-UCRs are involved in various aspects of neuroblastoma progression.

### 4.8. T-UCR Expression and Retinoic Acid Treatment

The differentiating agent all-trans-retinoic acid (ATRA) has been used to treat children with neuroblastoma; however, no information is available describing how T-UCRs play a role in neuroblastoma chemotherapy responses. Watters et al. investigated the differential regulation of T-UCR expression in three neuroblastoma cells (SH-SY5Y, SK-N-BE, and LAN-5) followed by ATRA-mediated differentiation for seven days [44], which resulted in differential expression of 32 T-UCR transcripts, including 16 up-regulated and 16 down-regulated in each of the cell lines. Two T-UCRs (uc.324 and uc.300A) were randomly chosen for validation experiments. The first, uc.324, was slightly up-regulated following treatment, whereas uc.300A was found to be down-regulated significantly following treatment. The knockdown of uc.300A leads to decreased cell viability due to the down-regulation of cell proliferation and cell invasiveness, suggesting a tumor-supportive role in neuroblastoma. These studies indicate that T-UCRs may play an essential role in mediating an ATRA-induced differentiation pathway, which could be through the regulation of T-UCR specific genes or miRNAs. Nevertheless, future studies will be needed to find the precise role of T-UCRs in cancer and normal cellular development.

## 5. Transcribed-Ultra Conserved Regions in Other Cancers

### 5.1. T-UCRs in Hepatocellular Carcinoma (HCC)

A group led by Patel et al. found an oncogenic role for uc.338 in hepatocellular carcinoma through the regulation of genes involved in transcription, cell cycle, ubiquitin cycle, and cell division [45]. This group also showed that uc.338 is also associated with a molecular functional classification, such as ligase activity, binding of proteins, nucleotides, and ATP, thus potentiating the cell growth in hepatocellular carcinoma (HCC) [45]. Furthermore, the presence of uc.339 in extracellular vesicles, such as exosomes and microvesicles, was discovered, providing a mechanism of uc.338 transfer from one cell to another within the tumor microenvironment, which could potentiate the progression of hepatocellular cancer [46]. This was the first study describing the presence of T-UCRs in exosomes [46]. Another study has shown that the Wnt pathway is involved in the progression of liver cancer through uc.158 [47]. In addition, during macrophage polarization, uc.306 was shown to be expressed in low levels, serving as a potential biomarker in hepatitis B-induced HCC [48].

### 5.2. T-UCRs in Bladder Cancer

Polycomb protein, Yin Yang 1 (YY1), mediates the interaction between uc.8 and miR-596 through protein-RNA binding, providing an additional layer of regulation in bladder cancer cells [49]. In a gastric cancer model, uc.416A acts as an oncogene by targeting miR-153 expression [23]. In bladder cancer, uc.8+ has been shown to be involved in bladder cancer progression through the stabilization of MMP9 by targeting miR-596 [20].

### 5.3. T-UCRs in Pancreatic, Lung, Prostate, and Breast Cancers

Activation of the oncogenic pathway by uc.190, uc.233, and uc.270, has been found in pancreatic adenocarcinoma [50]. Recently, we showed that uc.339 acts as an oncogene by decoying tumor suppressive miR-339-3p, miR-663b-3p, and miR-95-5p, leading to the activation of Cyclin E2 in non-small cell lung cancer patients [51]. Overexpression of uc.63 is involved in the progression of castration-resistant prostate cancer through MMP2 by regulating the expression of miR-130b [52]. In breast cancer patients (luminal A subtype), uc.63 promotes the survival of cancer cells [53].

### 5.4. T-UCRs in Gastric and Colon Cancers

Methylation in the promoter region was associated with lower levels of uc.160, suggesting a tumor suppressive role in gastric cancer [54]. Higher expression of uc.261 is implicated in damaging the lining of the digestive tract in Crohn’s disease, a chronic inflammatory bowel disease [55]. The tumor-suppressive activity of uc.160 was found through PTEN stabilization by targeting oncogenic miR-155 in gastric cancer [56]. By reducing primary miRNA-195 levels through direct binding, uc.173 has been shown to be involved in stimulating the renewal of intestinal epithelium [57]. Gut permeability is regulated by uc.173 by decreasing the levels of miR-29b that target claudin-1 mRNA [58]. Methylation of uc160 and uc346 was found in the plasma of colorectal cancer patients [59]. Studies have shown the diagnostic and prognostic potential of uc.73 and uc.388 in colorectal cancer patients [60].

### 5.5. T-UCRs in Other Cancers

One T-UCR, uc.283, is abundantly expressed in pluripotent embryonic stem cells, as well as in a variety of solid cancers, including gliomas [61]. A specific T-UCR signature (up-regulation of uc.58, uc.202, and uc.207, along with down-regulation of uc.214) is currently known to be involved in the progression of Barrett’s esophagus to esophageal adenocarcinoma [62]. The tumor suppressive role of uc.38, targeting pre-B-cell leukemia homeobox 1 (PBX1) protein, has been found in breast cancer [63]. Furthermore, T-UCRs were also involved in the development of the nervous system [64]. An upregulation of uc.416 has been found to be involved in promoting the epithelial-to-mesenchymal transition through miR-153 in renal cell carcinoma [65]. These studies suggest that T-UCRs are involved, not only in neuroblastoma, but also in other cancers.

## 6. Therapeutic Approaches in Targeting Transcribed-Ultra Conserved Regions

T-UCRs are involved in a wide variety of diseases, including, but not limited to, various solid cancers, hematological malignancies, neuronal development, Crohn’s disease, hepatitis B infection, and the renewal of intestinal epithelium. Therapeutic strategies targeting T-UCRs are limited at present; therefore, future studies are required to explore this exciting opportunity. In a vast majority of cancers, T-UCRs are found to be upregulated. There are several ways to target over-expressed oncogenic microRNAs, including the use of anti-sense oligonucleotides. Further research is required in order to investigate where a similar method may be applied to T-UCRs. Silencing RNAs (siRNA) are also involved in reducing the levels of long non-coding RNAs, not only in cancer, but also in other diseases such as Alzheimer’s disease. Approximately 20 clinical trials have been conducted using siRNA- and/or miRNA-based therapies, including one using a compound called SPC3649 that inhibits miRNA-122 function. There are more than 10 patents in the United States and European Union related to miRNAs and siRNAs [66]. Similar to miRNA sponges, T-UCRs can also be targeted. Finally, DNA methylation agents have been shown to restore the expression of tumor-suppressive T-UCRs in experimental systems and should be considered for another potential avenue.

## 7. Conclusions and Future Prospective

Dis-regulation of T-UCR expression is involved in the biology of neuroblastoma. Although the discovery of T-UCRs occurred over a decade ago, the progress on the functional aspects of cell homeostasis has remained limited. The role of T-UCRs in the normal development of the central nervous system has proved to be difficult to understand. Several mechanistic aspects on progression, metastasis, and therapy resistance by T-UCRs are missing in the literature. Recently, the presence of T-UCRs in exosomes has been discovered, but their potential role in influencing the tumor microenvironment is not yet known. Research on the role of T-UCRs in neuroblastoma is rapidly increasing. This unique relationship is essential to understanding the development of chemotherapeutic and immunotherapeutic resistance, as well as uncovering novel therapeutic avenues in neuroblastoma.

## Figures and Tables

**Figure 1 ncrna-05-00039-f001:**
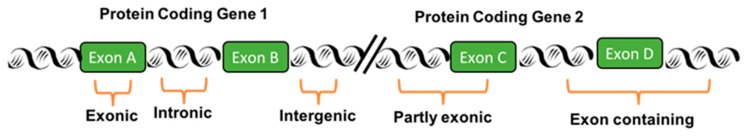
Types of ultra-conserved regions (UCRs). A schematic representation of the different types of UCRs as per their genomic location with respect to their protein-coding genes.

**Figure 2 ncrna-05-00039-f002:**
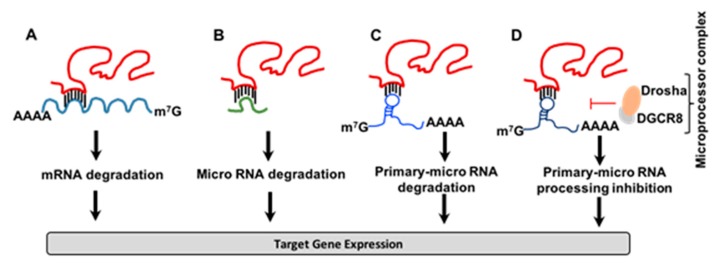
Regulation of gene expression by Transcribed-Ultra Conserved Regions (T-UCRs). T-UCRs regulate gene expression by (**A**) direct interaction with 3′UTR of specific mRNA, (**B**) trapping miR, (**C**) degradation of primary miR or (**D**) inhibition of primary miR processing mechanism by microprocessor complex.

**Figure 3 ncrna-05-00039-f003:**
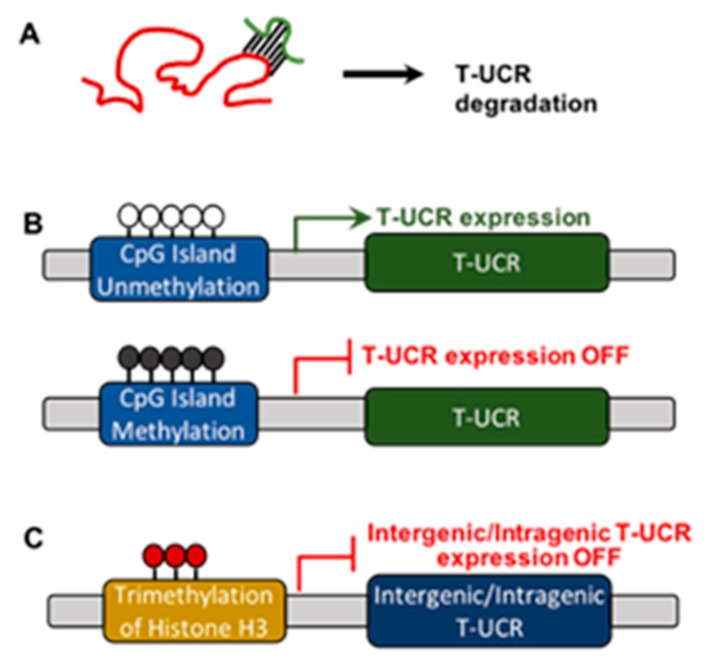
Regulation of T-UCR expression by (**A**) miR, (**B**) CpG island methylation, or (**C**) trimethylation of histone H3.

**Table 1 ncrna-05-00039-t001:** A list of T-UCRs negatively correlated with miRNAs in short- vs. long-term survivors of neuroblastoma.

T-UCR Name	Chromosome Location	Start (bp)	End (bp)	T-UCR Expression	miRNA	miRNA Expression
uc.209	7	23,561,888	23,562,137	↑	hsa-miR-877-3p	↓
uc.271	9	128,304,352	128,304,562	↑	hsa-miR-383	↓
uc.312	10	120,076,537	120,076,858	↑	hsa-miR-877-3p, hsa-miR-548d-5p	↓
uc.330	11	66,393,896	66,394,102	↑	hsa-miR-548d-5p	↓
uc.371	14	36,020,189	36,020,484	↑	hsa-miR-877-3p	↓
uc.411	17	35,329,619	35,329,847	↑	hsa-miR-33b-5p	↓
uc.421	18	22,693,155	22,693,499	↑	hsa-miR-877-3p	↓
uc.435	18	53,089,931	53,090,157	↑	hsa-miR-939	↓
uc.452	19	31,827,947	31,828,150	↑	hsa-miR-383	↓

T-UCR—Transcribed Ultra Conserved Region; bp—base pairs; hsa—Homo Sapiens; miR—microRNA; ↑—Upregulation; ↓—Downregulation.

**Table 2 ncrna-05-00039-t002:** A list of T-UCRs correlated with DNA copy number changes in neuroblastoma patients.

T-UCR	Location	Start (bp)	End (bp)
uc.10	1	10,965,574	10,965,848
uc.25	1	51,166,034	51,166,268
uc.300	10	102,547,118	102,547,325
uc.303	10	103,052,427	103,052,698
uc.308	10	103,245,812	103,246,088
uc.379	14	97,431,368	97,431,619
uc.380	14	97,762,594	97,762,825

T-UCR—Transcribed-Ultra Conserved Region; bp—base pairs.

**Table 3 ncrna-05-00039-t003:** A list of T-UCR clusters and the associated pathways in neuroblastoma patients.

Cluster	Pathway	T-UCR	Chromosome Location
Cluster 1		uc.31	1
		uc.58	2
	DNA	uc.130	3
	Damage	uc.139	4
	Response	uc.196	6
		uc.293	10
		uc.296	10
		uc.365	14
		uc.405	16
Cluster 2		uc.74	2
		uc.103	2
		uc.104	2
		uc.131	3
		uc.134	3
	Cell Cycle and	uc.257	9
	Proliferation	uc.277	9
		uc.278	9
		uc.279	9
		uc.431	18
		uc.444	19
		uc.483	3
Cluster 3		uc.16	1
		uc.30	1
		uc.46	1
		uc.49	2
	Differentiation	uc.101	2
		uc.193	6
		uc.366	14
		uc.380	14
		uc.456	20
Cluster 4		uc.21	1
		uc.65	2
	Immune	uc.98	2
	Response and	uc.145	4
	Development	uc.334	11
		uc.347	13

T-UCR—Transcribed-Ultra Conserved Region; bp—base pairs.

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
