# Peer review of "The Transcribed-Ultra Conserved Regions: Novel Non-Coding RNA Players in Neuroblastoma Progression"

_ncrna, 2019, doi:10.3390/ncrna5020039_

Round 1
Reviewer 1 Report
In the manuscript entitled ‘The transcribed unltraconserved regions: Novel noncoding RNA players in neuroblastoma progression and clinical implications’ by Shaw et al, the authors attempt to provide an overview of the existing literature regarding the role of T-UCRs in neuroblastoma. While the authors did try to largely cover the current knowledge, the manuscript is merely a list of superficially discussed papers and lacks critical thinking, prioritization of the importance of the available reports and constructive synthesis and digestion of all the cited evidence. Additionally, the quality of the language in the manuscript is particularly low, which renders reading and understanding the authors’ argumentation not trivial.
More specifically:
1. The title of the manuscript suggests that the clinical implications of T-UCRs in neuroblastoma will be evenly discussed in the text. However, there is only a very small part at the end of the manuscript dedicated to this. Moreover, this particular part is only speculative, lacks any relevant data and therefore also any pertinent literature references.
2. The authors should discuss the existing literature more critically. More information on the systems used in each of the discussed studies should be provided, was it a rodent-only study, what was the sample size used, have the data been replicated by independent research groups etc.. Moreover, the authors should highlight the limitations of each of the papers they discuss.
3. The argument that ‘the scientific community is ignoring the 98% of the genome (junk DNA)’ is wrong at many different levels. A. the percentage of genomic dark matter is no longer considered to be 98%; this information is outdated. B. There is massive research performed on the noncoding part of the genome in (at least) the last 20 years. The fact that the authors are not aware of it does not mean that the scientific community as a whole ignores noncoding DNA.
4. Related to Figure 1: ‘sense’ and ‘antisense’ are defined as ‘in the same’ or ‘opposite direction’. What’s the reference as to what is ‘same’ or ‘opposite’? The authors need to be more explicit for non familiar readership.
5. The section #3 also includes a part on T-UCR-induced mechanisms which is not indicated by the title of the section.
6. Several sections/figures are solely based on 1 single paper (e.g. sections 4….). Obviously, the available evidence is not adequate to support the authors’ claims. Moreover, if only one paper is available on a certain subject then this subject does not obviously merit a separate section.
The manuscript could massively profit from some proof-reading, since there are too many typos, grammatical and conceptual mistakes. I only highlight here few examples: exosmic ncRNAs, MYCH, feature prospectives….
Author Response
Response to Reviewer #1:
In the manuscript entitled ‘The transcribed ultraconserved regions: Novel noncoding RNA players in neuroblastoma progression and clinical implications’ by Shaw et al, the authors attempt to provide an overview of the existing literature regarding the role of T-UCRs in neuroblastoma. While the authors did try to largely cover the current knowledge, the manuscript is merely a list of superficially discussed papers and lacks critical thinking, prioritization of the importance of the available reports and constructive synthesis and digestion of all the cited evidence. Additionally, the quality of the language in the manuscript is particularly low, which renders reading and understanding the authors’ argumentation not trivial.
We deeply appreciate this reviewer’s comments and scientific criticism that have greatly improved our manuscript. We have enhanced the quality of the language by performing edits at our institutions editing services.
Major comments:
1. The title of the manuscript suggests that the clinical implications of T-UCRs in neuroblastoma will be evenly discussed in the text. However, there is only a very small part at the end of the manuscript dedicated to this. Moreover, this particular part is only speculative, lacks any relevant data and therefore also any pertinent literature references.
My response: This is indeed an important question. T-UCRs involvement in neuroblastoma is a budding area of investigation. So far a few studies have focused on the role of T-UCRs in the progression of neuroblastoma. Therefore, the clinical implications have been deleted from the title, and now, the new title should read as: The Transcribed-ultra conserved regions: Novel non-coding RNA players in neuroblastoma progression
2. The authors should discuss the existing literature more critically. More information on the systems used in each of the discussed studies should be provided, was it a rodent-only study, what was the sample size used, have the data been replicated by independent research groups etc. Moreover, the authors should highlight the limitations of each of the papers they discuss.
My response: This is also a vital issue and we appreciate the reviewers’ concern. As suggested by the reviewer, we have now included the information in the revised manuscript.
3. The argument that ‘the scientific community is ignoring the 98% of the genome (junk DNA)’ is wrong at many different levels. A. the percentage of genomic dark matter is no longer considered to be 98%; this information is outdated. B. There is massive research performed on the noncoding part of the genome in (at least) the last 20 years. The fact that the authors are not aware of it does not mean that the scientific community as a whole ignores noncoding DNA.
My response: This is logical comment and we welcome the reviewers’ analyses. We are in complete agreement with the reviewer and have now changed the tone of the statement. Futhermore, we discussed recent findings reported in Cell, 2018, 172 (3)491-499 entitled Ultraconserved Enhancers Are Required for Normal Development in the revised manuscript.
4. Related to Figure 1: ‘sense’ and ‘antisense’ are defined as ‘in the same’ or ‘opposite direction’. What’s the reference as to what is ‘same’ or ‘opposite’? The authors need to be more explicit for non-familiar readership.
My response: We thank the reviewer for this critical comment. As suggested, we have added the statement, “If the T-UCRs are transcribed towards the host gene, called sense direction whereas opposite direction of the host gene called anti-sense direction,” in the revised manuscript.
5. The section #3 also includes a part on T-UCR-induced mechanisms which is not indicated by the title of the section.
My response: We appreciate the reviewer for identifying and highlighting this issue. As suggested, we added subtitles to each of the subsections in the revised manuscript.
6. Several sections/figures are solely based on 1 single paper (e.g. sections 4....). Obviously, the available evidence is not adequate to support the authors’ claims. Moreover, if only one paper is available on a certain subject then this subject does not obviously merit a separate section. The manuscript could massively profit from some proof-reading, since there are too many typos, grammatical and conceptual mistakes. I only highlight here few examples: exosmic ncRNAs, MYCH, feature prospective.
My response: We are grateful to the reviewer for calling attention to these imperative errors. Dissecting the role of T-UCRs in neuroblastoma progression is a budding area of investigation. In total, five articles were published on neuroblastoma. Since the present special issue focus on “Non-Coding RNA and Brain Tumors,” we have prepared separate sections on neuroblastoma based on the biological process. Moreover, we also provided a brief review of T-UCRs in other human’s cancers as well. We have modified this section as advised by the reviewer, and we have edited our revised manuscript through our institutional editing services.
Reviewer 2 Report
This manuscript by Shaw and colleagues is an interesting review of the role that transcribed ultra-conserved regions may play in neuroblastoma (and other cancers). I especially appreciate the authorship given to an REU student. This manuscript covers some useful material for other scientists in the field, however there are some grammatical issues that need correcting before it should be published. I have listed some of them below but in general, several of the paragraphs could use with a more unified structure (broad introductory sentence, material related to that sentence, and then a summary, concluding statement). Some of the paragraphs read like a laundry list which will bog the intended audience down.
More “major” comments:
I don’t understand how Table 2 reflects changes in copy number impacting gene expression. In fact, this section (4.5) is not very well fleshed out and confusing. Is the genome being duplicated, are these fragments being duplicated (but not the genome), or are these T-UCRs actually highly sequence similar to one another?
End of section 4.6 needs a better concluding statement. It seems like the paragraph just ends without a conclusion. Are the 40 T-UCRs that are p53 responsive (this sentence needs grammar editing) thought to be activating p53, involved in the pathway, or otherwise?
Maybe break up section 5 into a couple of paragraphs that follow a better arch. In addition the section needs a better overall introduction and conclusion to keep it from being a laundry list of examples of T-UCRs in other cancers.
This is the same for section 4.8 (needs a better concluding sentence).
If section 4.7 is referring to published results from Bejerano et al (#12) is it really necessary to have a table with the coordinates for the T-UCRs associated with each cluster? I feel like the tables with the UCR coordinates are a little unnecessary in the main text when the original reference is listed. If the authors felt it necessary, perhaps a supplemental table listing all of the T-UCRs in a friendly to use tab-delimited file? Just to clarify, I am suggesting moving the coordinates to a different, supplemental table, but leaving the necessary information (such as table 1) in the body of the text.
Please pay attention to the tenses used throughout the document, particularly when describing work performed in the past. Some statements like “has found” or “have found” can be replaced with “found”. (lines 106, 111), whereas others are reflecting roles of T-UCRs that are still ongoing, such as line 271, where it should say “T-UCRs are involved in a wide…”.
Minor comments:
How can a T-UCR be sense or antisense (lines 61-66)? Is this referring to T-UCRs that overlap with a protein-coding gene? A conserved region is inherently conserved on both strands, so it doesn’t make sense that there would be a directionality to them aside from transcription itself.
Reference 39 is cited on line 219. How does this reference have to do with T-UCRs?
Line 217 “Custer” should be “Cluster”
Line 204, provide the abbreviation for Gene Set Enrichment Analysis (GSEA) here, as this abbreviation is used later on in line 214.
Line 202, referring to “authors” – what authors?
Line 46, is “MYCH” a typo? If it is supposed to reflect MYCN, what does MYCN itself stand for?
Line 285: “Feature Prospective” should be “Future Prospective”
Line 283 “have been proved” should be “have been proven”
Line 277 “Silencing (Si) RNAs” is not actually the definition of siRNA. This should instead be “small interfering RNAs”.
Author Response
Response to Reviewer #2:
This manuscript by Shaw and colleagues is an interesting review of the role that transcribed ultra-conserved regions may play in neuroblastoma (and other cancers). I especially appreciate the authorship given to an REU student. This manuscript covers some useful material for other scientists in the field, however there are some grammatical issues that need correcting before it should be published. I have listed some of them below but in general, several of the paragraphs could use with a more unified structure (broad introductory sentence, material related to that sentence, and then a summary, concluding statement). Some of the paragraphs read like a laundry list which will bog the intended audience down.
My response: We thank the reviewer for bringing light to this interesting review article that covers useful information for the scientific community in the field of neuroblastoma. Additionally, summer undergraduate students are involved in this project and deserve authorship. We appreciate the reviewer’s comment on the student authorship. As recommended by the reviewer, we have revised this manuscript and the grammatical errors have been corrected by our institutional editing services.
Major comments:
1) I don’t understand how Table 2 reflects changes in copy number impacting gene expression. In fact, this section (4.5) is not very well fleshed out and confusing. Is the genome being duplicated, are these fragments being duplicated (but not the genome), or are these T-UCRs actually highly sequence similar to one another?
My response: We are grateful to this reviewer for his critical analysis. The T-UCRs which are linked to the DNA copy number changes in neuroblastoma patients are given in Table 2. As suggested by the reviewer, we have modified section 4.5. This is about the duplication of UCR sequences in the genome.
2) End of section 4.6 needs a better concluding statement. It seems like the paragraph just ends without a conclusion. Are the 40 T-UCRs that are p53 responsive (this sentence needs grammar editing) thought to be activating p53, involved in the pathway, or otherwise?
My response: We are grateful for this reviewer’s recommendation, and we have amended section 4.6 to have a better concluding statement as proposed.
3) Maybe break up section 5 into a couple of paragraphs that follow a better arch. In addition, the section needs a better overall introduction and conclusion to keep it from being a laundry list of examples of T-UCRs in other cancers.
My response:. The suggestion posed by the reviewer is greatly appreciated, as it improves the flow of section 5 of the manuscript. The section is modified in the revised manuscript as advised.
4) This is the same for section 4.8 (needs a better concluding sentence). If section 4.7 is referring to published results from Bejerano et al (#12) is it really necessary to have a table with the coordinates for the T-UCRs associated with each cluster? I feel like the tables with the UCR coordinates are a little unnecessary in the main text when the original reference is listed. If the authors felt it necessary, perhaps a supplemental table listing all of the T-UCRs in a friendly to use tab-delimited file? Just to clarify, I am suggesting moving the coordinates to a different, supplemental table, but leaving the necessary information (such as table 1) in the body of the text.
My response: These are excellent points raised by the reviewer. Section 4.8 is modified, and the information related to UCR coordinates from Table 4 has been deleted as recommended by the reviewer.
5) Please pay attention to the tenses used throughout the document, particularly when describing work performed in the past. Some statements like “has found” or “have found” can be replaced with “found”. (lines 106, 111), whereas others are reflecting roles of T-UCRs that are still ongoing, such as line 271, where it should say “T-UCRs are involved in a wide...”.
My response: We appreciate the suggestions offered by the reviewer and their explanation of the grammatical errors present throughout the manuscript. These errors were examined and amended as encouraged.
Minor comments:
1. How can a T-UCR be sense or antisense (lines 61-66)? Is this referring to T-UCRs that overlap with a protein-coding gene? A conserved region is inherently conserved on both strands, so it doesn’t make sense that there would be a directionality to them aside from transcription itself.
My response: We thank the reviewer for pointing out this issue. The lines are modified in the revised manuscript, and the clarity of this statement was improved after using this reviewer’s advice.
2. Reference 39 is cited on line 219. How does this reference have to do with T-UCRs?
My response: The reference in question has been corrected.
3. Line 217 “Custer” should be “Cluster”
My response: The spelling error has been amended.
Round 2
Reviewer 1 Report
The authors have addressed all my major concerns.